# Peer review of "Expert Clinical Management of Severe Immune-Related Adverse Events: Results from a Multicenter Survey on Hot Topics for Management"

_jcm, 2022, doi:10.3390/jcm11205977_

Round 1

Reviewer 1 Report

In the present manuscript “Expert clinical management of severe immune-related adverse 2 events: results from a multicenter survey on hot topics for management”, the authors approach the topic of differences in guideline recommendations for irAEs management: As solid evidence on irAEs management is scarce, most recommendations are based on expert opinions. This issue is of particular clinical relevance in case of grade III/IV irAEs as these events are rare and evidence is therefore even harder to obtain. To highlight this issue, the authors created three surveys on hepatitis, renal and mysositis- related grade III irAEs, which were sent to selected experts in the field and appear to reveal substantial management differences.

Whereas the authors approach a clinically relevant topic by creative means, trying to highlight respective guideline limitations by providing descriptive statistics, the applied methodology raises queries.

First, the actual aim of the present paper is not clearly stated. I may assume the authors hypothesized that differences in guideline recommendations lead to differences in “real life” management? Therefore, the goal would have been be to highlight these limitations (and therefore call for a homogenization of recommendations, or more data to allow for guideline homogenization?) I would ask to clarify the “mission statement” and clinical relevance of the present survey in both the abstract and the introduction.

Second, as for all surveys, limited data quality alleviates direct clinical value. As long as no better evidence is available, surveys may provide the only source to accumulate evidence (or i.e. allow to demand guideline adaptations). If so, however, authors are obliged to minimize bias by providing solid information on applied methods.  

-          I may ask to clarify how and why survey participants where exactly selected. The manuscript states:

“All participants were physicians selected on the basis of their knowledge and clinical practice in each of the 3 topics, and on their academic credentials in the area on the basis of publications and participation in research activities related to irAEs.”

Why were only 119 experts approached? How was the selection process performed? Why not send the surveys eg. to every university hospital in the EU + US or to all ASCO- members and assess the answers stratified by the number of patients treated by year? This approach would even allow for more power and, most likely, exploratory statistics. As I assume the authors had a rationale and a standardized selection process, I may ask to provide the information in the manuscript.

Line 162ff: “We surveyed participants from two groups conducting research in irAE in rheu- 162 matology: Outcome Measures in Rheumatology (OMERACT) immune-related adverse 163 events Special Interest Group [7], and a newly formed consortium in the US interested in 164 collecting prospective data in cancer patients who develop irAE with ICI.” – for one of the three surveys the requested information is eg. provided in the results section. This information is supposed to be explained in the methods section.

-          I may ask for the criteria based on which the survey was designed. The manuscript states:

“Survey participants for each of the three areas of interest were selected by the team author for each area”

I may therefore request the authors to provide information how and why the surveys where designed the present way . Do they follow any guidelines for reporting survey-based research?

Finally, two minor remarks:

1-     Introduction line 42: ~ there are over 20 indications for ICI therapy in cancer. I may ask to either quote the line or clarify it - what are “indications”? By means of FDA-approvals, I would assume there are way more than 20 as I would spontaneously recall almost 10 for gynecologic malignancies only.

2-    The methods state that SPSS v. 26 was used to perform all analyses. To me, figure 1 rather appears like Microsoft Excel? I may be mistaken, but I may ask for clarification. Please correct the punctuation in the figure (English “,” -> “.”)

Author Response

We would like to thank the reviewers for their thoughtful review of our manuscript which we have now revised according to their suggestions. We have also had a scientific editor review the manuscript to correct/improve any language or grammar issues.

Reviewer #1

  1. In the present manuscript “Expert clinical management of severe immune-related adverse events: results from a multicenter survey on hot topics for management”, the authors approach the topic of differences in guideline recommendations for irAEs management: As solid evidence on irAEs management is scarce, most recommendations are based on expert opinions. This issue is of particular clinical relevance in case of grade III/IV irAEs as these events are rare and evidence is therefore even harder to obtain. To highlight this issue, the authors created three surveys on hepatitis, renal and mysositis- related grade III irAEs, which were sent to selected experts in the field and appear to reveal substantial management differences.

Whereas the authors approach a clinically relevant topic by creative means, trying to highlight respective guideline limitations by providing descriptive statistics, the applied methodology raises queries.

First, the actual aim of the present paper is not clearly stated. I may assume the authors hypothesized that differences in guideline recommendations lead to differences in “real life” management? Therefore, the goal would have been be to highlight these limitations (and therefore call for a homogenization of recommendations, or more data to allow for guideline homogenization?) I would ask to clarify the “mission statement” and clinical relevance of the present survey in both the abstract and the introduction.

We appreciate the reviewer’s comment. Since there is no hard evidence regarding the management of ICI-related adverse events, especially in the case of severe (grade III/IV) events, we developed surveys focused on three commonly affected organs (liver, kidney and muscle) which have high risk for morbidity, sequelae, and mortality. Our aim was to evaluate expert-driven real-world management of these patients. This aim has been highlighted in both the Abstract and Methods section (pages 1 and 2).

  1. Second, as for all surveys, limited data quality alleviates direct clinical value. As long as no better evidence is available, surveys may provide the only source to accumulate evidence (or i.e. allow to demand guideline adaptations). If so, however, authors are obliged to minimize bias by providing solid information on applied methods.

We agree with the reviewer and, in fact, the scarce data about some important practice aspects such as indication and timing for biopsy or therapy with corticosteroids and other agents for patients with severe irAEs, were the main reasons for conducting this study. We have now added additional information on participants. The actual surveys on ICI-induced hepatitis, AKI and myositis are provided in the appendix of the manuscript (pages 12-16).

  1. I may ask to clarify how and why survey participants where exactly selected. The manuscript states: “All participants were physicians selected on the basis of their knowledge and clinical practice in each of the 3 topics, and on their academic credentials in the area on the basis of publications and participation in research activities related to irAEs.”

The following has been added “Participants were selected if they met the following criteria: active practice in patients with irAEs and expertise based on the authorship of at least 1 published article on the topic of interest. In the case of myositis the contacted experts were members of either the OMERACT (Outcome Measures in Rheumatology) irAE initiative or a newly formed consortium in the United States investigating irAEs. In the case of acute kidney injury the contacted experts were members of the ICI-Acute Kidney Injury (AKI) Consortium Investigators in page 2.

  1. Why were only 119 experts approached? How was the selection process performed? Why not send the surveys eg. to every university hospital in the EU + US or to all ASCO- members and assess the answers stratified by the number of patients treated by year? This approach would even allow for more power and, most likely, exploratory statistics. As I assume the authors had a rationale and a standardized selection process, I may ask to provide the information in the manuscript.

The aim of the study was not so much to ascertain how oncologists or other specialists practice at large, which we assume would show large differences, but rather to evaluate how a select group of experts with extensive experience in the field manage these irAEs. Our findings show that even among experts there is little consensus for important decisions, even despite guidelines, highlighting the need for additional research in this important field. This has been added to the discussion.

  1. Line 162ff: “We surveyed participants from two groups conducting research in irAE in rheumatology: Outcome Measures in Rheumatology (OMERACT) immune-related adverse events Special Interest Group [7], and a newly formed consortium in the US interested in collecting prospective data in cancer patients who develop irAE with ICI.”

–     for one of the three surveys the requested information is eg. provided in the results section. This information is supposed to be explained in the methods section.

The three surveys (liver, AKI and myositis) are available at the appendix section.

  1. I may ask for the criteria based on which the survey was designed. The manuscript states: “Survey participants for each of the three areas of interest were selected by the team author for each area”

I may therefore request the authors to provide information how and why the surveys where designed the present way. Do they follow any guidelines for reporting survey-based research?

The surveys were developed by the authors on the basis of topics considered to be most controversial and clinically relevant for the diagnosis and treatment of these irAEs.

  1. Introduction line 42: ~ there are over 20 indications for ICI therapy in cancer. I may ask to either quote the line or clarify it - what are “indications”? By means of FDA-approvals, I would assume there are way more than 20 as I would spontaneously recall almost 10 for gynecologic malignancies only.

Modified to “there are numerous approved indications for ICI therapy across many cancers” and a reference has been added (see page 2).

  1. The methods state that SPSS v. 26 was used to perform all analyses. To me, figure 1 rather appears like Microsoft Excel? I may be mistaken, but I may ask for clarification. Please correct the punctuation in the figure (English “,” -> “.”)

SPSS was used to perform the statistical analysis but, as the reviewer has pointed out, the figure was performed by Excel format. This information has been now been added to the new version of the manuscript in the Methods section (page 2). The figure has been modified without decimals to improve readability.

Reviewer 2 Report

This is a merely descriptive study about the clinical practice on some immune related adverse effects of ICI as cancer therapy. The article is well written, but the knowledge deriving from this study is limited because of small number of participants, just 3 types of toxicity are considered and cancer types are not specified in the survey. However, I suggest:

-       In the introduction section a more extensive explanation of these immune-related toxicities should be provided.

-       The results in paragraph 3.2 and 3.3 should also be summarized in graphs.

Author Response

We would like to thank the reviewers for their thoughtful review of our manuscript which we have now revised according to their suggestions. We have also had a scientific editor review the manuscript to correct/improve any language or grammar issues.

Reviewer #2

  1. This is a merely descriptive study about the clinical practice on some immune related adverse effects of ICI as cancer therapy. The article is well written, but the knowledge deriving from this study is limited because of small number of participants, just 3 types of toxicity are considered and cancer types are not specified in the survey. However, I suggest:

In the introduction section a more extensive explanation of these immune-related toxicities should be provided.

A new paragraph explaining the importance of liver, kidney and muscle adverse events has been added to the introduction, as well as two new references (ref. 6 and 7).

  1. The results in paragraph 3.2 and 3.3 should also be summarized in graphs.

Thank you for your suggestions, we have now included an additional graph on the treatment of myositis with myocarditis. As the data on biopsies was very simple and dichotomous we have left it in the text as we felt an additional figure would not be very informative.

Round 2

Reviewer 1 Report

I thank the authors for the clear answers and additional information. No further questions - I may recommend to accept the manuscript in its present form.